# Association between Practising Religion and Cardiovascular Disease Risk among Ghanaian Non-Migrants and Migrants in Europe: The RODAM Study

**DOI:** 10.3390/ijerph18052451

**Published:** 2021-03-02

**Authors:** Jessica Michgelsen, Daniel Boateng, Karlijn A.C. Meeks, Erik Beune, Juliet Addo, Silver Bahendeka, Karien Stronks, Charles Agyemang

**Affiliations:** 1Department of Public Health, Amsterdam UMC, University of Amsterdam, Amsterdam Public Health Research Institute, 1105 AZ Amsterdam, The Netherlands; k.a.meeks@amsterdamumc.nl (K.A.C.M.); e.j.beune@amsterdamumc.nl (E.B.); k.stronks@amsterdamumc.nl (K.S.); c.o.agyemang@amsterdamumc.nl (C.A.); 2Julius Global Health, Julius Center for Health Sciences and Primary Care, University Medical Center Utrecht, Utrecht University, 3584 CX Utrecht, The Netherlands; d.boateng-2@umcutrecht.nl; 3Center for Research on Genomics and Global Health, National Human Genome Research Institute, National Institutes of Health, Bethesda, MD 20892-5635, USA; 4Department of Non-Communicable Disease Epidemiology, London School of Hygiene and Tropical Medicine, London WC1E 7HT, UK; juliet.x.addo@gsk.com; 5MKPGMS–Uganda Martyrs University, Kampala 2227+XW, Uganda; Bahendeka@yahoo.com

**Keywords:** religion, cardiovascular diseases, transients and migrants, Africa south of the Sahara, Europe

## Abstract

(1) Background: Sub-Saharan African migrants residing in high-income countries are more affected by cardiovascular diseases (CVDs) and associated risk factors than host populations for unclear reasons. The aim was to explore the associations of religion and religious affiliations with CVD risk among Ghanaian non-migrants and migrants in Europe. (2) Methods: The 10-year CVD risk was estimated using pooled cohort equations for 3004 participants from the cross-sectional Research on Obesity and Diabetes among African Migrants (RODAM) study. Logistic regression analyses were conducted to assess associations between religion and elevated CVD risk (score ≥ 7.5) with adjustment for covariates. (3) Results: Religious men in Europe had a lower 10-year CVD risk compared with non-religious men (adjusted OR 0.51; 95% confidence interval 0.30–0.85), specifically men affiliated with Seventh-Day Adventism (0.24; 0.11–0.53) followed by other affiliations (0.32; 0.11–0.94) and Roman Catholicism (0.42; 0.21–0.86). The opposite was found in Ghana, with religious women having higher odds for elevated 10-year CVD risk (1.53; 1.02–2.30) compared with their non-religious counterparts, specifically women affiliated with Reformed Christianity (1.73; 1.03–2.90) and other denominations (2.81; 1.20–6.54). Associations were not significant for men in Ghana and women in Europe. Adjustments for social support, stress, and health behaviors did not meaningfully alter the associations. (4) Conclusions: Christian religious Ghanaian men living in Europe seem to have lower CVD risk compared with their non-religious counterparts, while Christian religious women in Ghana appear to have increased CVD risk. Further unravelling the contributing factors and the differences between sex and environmental settings is needed.

## 1. Introduction

Cardiovascular diseases (CVDs) currently account for one in three deaths worldwide [1,2]. Around 80% of these deaths occur in low- and middle-income countries (LMICs) bringing a double burden of diseases to health systems that are already overstretched by communicable diseases [3]. The risk of developing CVDs is highly influenced by lifestyle-related factors such as a high consumption of unhealthy food, physical inactivity, tobacco use, and harmful use of alcohol. These disadvantageous health behaviors are becoming more prevalent in LMICs due to elevated exposures and limited preventive services [4].

Migrant populations residing in high-income countries seem to be more affected by CVDs and associated risk factors than host populations [5]. The Research on Obesity and Diabetes among African Migrants (RODAM) study reported higher prevalence of elevated CVD risk (score ≥ 7.5%) for Ghanaian men living in Amsterdam (53.9%) and Berlin (61%) compared with rural Ghana (34.7%) [6]. Their CVD risk increased with a longer stay in Europe [6]. Other studies also found high prevalence of CVD risk among Sub-Saharan African (SSA) migrants compared to non-migrants [7,8]. Geographical differences among a relatively homogenous population indicate that genetic factors alone cannot account for elevated CVD risk and emphasise the role of other factors in the development of CVDs [9].

Religious and spiritual (R/S) affiliations may be an underlying factor for CVD risk variations. R/S affiliations play an important role in lives of SSA (migrant) populations with the majority being affiliated with Christianity (63%) [10]. These beliefs and related practices are known to shape one’s view of the world and behavioral patterns [11] and therefore have potential to both positively and negatively affect one’s health [12]. Positive aspects include encouragement to live healthily, provision of mechanisms to regulate stress, and contribution to building a social support network [12]. As such, evidence suggests an association between R/S and CVDs among e.g., European populations [13,14]. This could be the result of discouragement of alcohol consumption and smoking [14] or encouragement by religious leaders to be physically active [15]. On the contrary, unfavourable interactions with fellow church members and religious doubts or discontent can induce increased psychological stress levels, which is a known risk factor for CVDs [12,16].

Adherence to religion may change as a consequence of migration [17,18]. In times of separation migrants may rediscover their religion in which religious communities have abilities to offer support and intimacy. Or, in secular countries, migrants can experience change in religious behaviors or even complete religious alienation [17,18], which may be attributed to prioritisation of other activities such as work or getting familiar with the new culture. So far, the role of religion in CVD risk among African migrants living in a European context is unknown. Data on the role of religion on CVD risk are also lacking for non-migrants in Africa. Therefore, this study’s objective was to explore associations of religion and religious affiliations with CVD risk among Ghanaian non-migrants residing in Ghana and Ghanaian migrants in Amsterdam, Berlin, and London. We hypothesised different associations of practising religion and CVD risk between residents of Ghana and Ghanaian migrants in Europe due to differential exposure to contextual factors.

### Theoretical Framework

The theoretical framework in Figure 1 is partly based on two previously published models [19,20] and shows potential pathways between religion and CVD risk. As such, migration can influence one’s religious practices as religious alienation may be experienced by migrants in secular countries or religious communities may provide support in times of separation [17,18]. Practising religion can influence one’s social support network and behavioral self-regulation leading to particular health behaviors. These health behaviors can influence cardiometabolic conditions and subsequently CVD risk. Spirituality is more related to one’s emotional self-regulation, which could lead to less stress and therefore lower chances of developing cardiometabolic conditions and CVDs. Support from social networks can also influence this emotional self-regulation.

## 2. Methods

### 2.1. Study Design and Population

We used data from the multi-centre cross-sectional RODAM study of which the rationale and design have been described in detail elsewhere [9]. Data were collected between 2012 and 2015. Ghanaians were eligible to participate if aged between 25 and 70 years and living in rural or urban Ghana, or a first-generation migrant in Amsterdam, Berlin or London.

Recruitment of participants in Ghana was carried out in two purposely-chosen cities and fifteen randomly selected villages in the Ashanti region. Response rates were, respectively, 76% and 74% in rural and urban Ghana. In Amsterdam 67% of the randomly drawn individuals from Municipal Health registers responded to invitations, 53% of whom agreed to participate. In Berlin and London, participants were recruited through Ghanaian organisations and response rates were, respectively, 68% and 75%. Altogether, 6385 Ghanaians participated by completing a structured health questionnaire 5898 of whom were physically examined. Ethical approval was obtained from local ethics committees at all study sites and all participants gave informed written consent.

### 2.2. Measurements

Structured questionnaires provided information on demographics, length of stay in a respective European country, and whether participants practised religion (no, yes). When answering yes, follow-up questions included current religious denomination (no religion, Roman Catholic, Orthodox Christian, Protestant, Reformed Christian, Pentecostal, Seventh-Day Adventist (SDA), Islamic, or other) and frequency of attending religious services in past 6 months (once a week or more, once every 2 weeks, once a month, less than once a month, or never). This was categorised into once a week or more, once every 2 weeks or once a month, or less than once a month or never.

Based on the previously described theoretical framework, the following variables were chosen to be included in the analysis. Socioeconomic indicators (education and employment) were taken into account as potential confounders given their association with religious involvement and CVD risk [21]. Education was categorised into never been to school or elementary schooling only, lower vocational/secondary schooling, intermediate vocational/higher secondary schooling, or higher vocational schooling or university. Employment status included full-time employed, part-time employed, unemployed and looking for work, or other. Covariates were selected based on this paper’s theoretical model and comprised perceived social support, psychosocial stress, and health behaviors. Social support was based on the sum-score of five items (range 5–20) [22]. Stress-related variables included experience of a negative life-event in the past 12 months (yes or no) and stress at home and work (feeling irritable, filled with anxiety, or sleeping difficulties due to conditions at work or home) [23]. Levels of stress included questions about stress-situations at home and work (never, some periods, several periods or constantly). Answers to both questions were combined into a general stress score: Never experiences stress, experienced some periods of stress at work/home, experienced several periods of stress at home/work, or experienced permanent stress at home/work [23]. Relevant health behaviors included smoking status, daily energy intake, alcohol consumption, and physical activity. Smoking status was assessed positive if answered “yes” to the question “Do you smoke at all?”. Energy intake was calculated using the latest versions of the West African Food Composition Table and the German Nutrient Database (BLS 3.01, 2010) [24]. Alcohol beverages were assessed as number of units of beer, wine, and liquor per day [24]. Self-reported physical activity was categorised into low, medium, or high levels [25]. 

Fasting venous blood samples were collected, manually processed and immediately aliquoted, and then temporarily stored at the local research location at −20 °C. The samples were then transported to the respective local laboratories for registration and storage at −80 °C and were subsequently transported to Berlin, Germany, for biochemical analysis to avoid intra-laboratory variability. Serum total cholesterol, serum HDL-cholesterol, serum LDL-cholesterol, and fasting glucose levels were determined using ABX Pentra 400 chemistry analyzer (HORIBA ABX, Montpellier, France). Concentrations of total cholesterol and HDL-cholesterol were assessed using colorimetric test kits. Fasting plasma glucose concentration was measured using hexokinase. The definition of type 2 diabetes was based on World Health Organization (WHO) diagnostic criteria of fasting glucose ≥7.0 mmol/L, prescribed type-2-diabetes medication use, or self-reported diabetes [9]. Blood pressure (BP) was measured three times after five minutes of rest, using a validated semiautomatic device (Microlife WatchBP home). The average of the two final measurements was used in the analysis.

### 2.3. 10-Year CVD Risk

CVD risk score was calculated using the Pooled-Cohort-Equations (PCE) risk calculator developed for different ethnic groups [26]. The PCE predicts a 10-year risk of developing hard atherosclerotic cardiovascular diseases (ASCVD)—nonfatal myocardial infarction, fatal coronary heart disease, and (non)fatal stroke. The PCE was developed and validated among African American and European and Asian men and women, and it has been shown to be comparatively more precise in estimating CVD risk [27]. The equations are derived, using pooled data from ethnically and geographically diverse community-based cohorts, permitting the creation of sex- and ethnic-specific equations for non-Hispanic White American and African American women and men [26]. The equation combines age, sex, total cholesterol, HDL-cholesterol, systolic BP, diagnosis of type 2 diabetes, smoking, and usage of antihypertensive medication into estimations of absolute 10-year CVD risk (referred to as CVD risk in following sections). CVD risk has been categorised into low risk (<7.5%) and elevated risk (≥7.5%) [6]. The distribution of estimated 10-year CVD risk between Ghanaian migrants and home populations has been shown in previous analyses [28].

### 2.4. Data Analysis

Data were analysed using STATA version 14.2. CVD risk was estimated for RODAM participants aged 40–70 with total-cholesterol ranging from 130–320 mg/dL, HDL-cholesterol 20–100 mg/dL, systolic BP 90–200 mmHg, and without prior history of stroke and heart attack (*n* = 3124) [26]. Participants with missing data for practising religion (*n* = 39), religious denominations (*n* = 24), or frequency of attending religious services (*n* = 57) were excluded leaving a total number of 3004 participants (Appendix A). Participant characteristics were summarised as means with standard deviation, medians with 25th–75th percentiles, or numbers with percentages. Binary logistic regressions were used to assess associations between practising religion and odds of elevated CVD risk with adjustment for education, employment, social support, psychosocial stress, health behaviors, and length of stay in Europe. Age and smoking status were not included as intermediary factors because these were included in the CVD risk calculation. Models were stratified by sex and recruitment site. Similar analyses were conducted to determine associations of religious denominations and frequency of attending religious services with CVD risk. Ghanaian women in Europe, practising Islam, were omitted from religious denominations analysis due to a too small sample size (*n* = 8). A subset analysis was conducted to examine whether associations between practising religion and CVD risk varied in rural or urban Ghana (for non-migrants) and in Amsterdam, Berlin, or London (for migrants). Statistical significance level was set at *p* < 0.05 for all tests.

## 3. Results

### 3.1. General Characteristics

Table 1 summarises characteristics of male participants stratified by site and practising religion or not. Among men, 69.8% (*n* = 273) in Ghana practised religion compared to 83.4% (*n* = 649) in Europe. Smoking and alcohol use were higher among non-religious men compared with those practising religion in both Ghana and Europe. Men not practising religion reported higher physical activity levels than men practising religion. Slightly fewer type-2 diabetes cases were detected among men practising religion in Ghana and Europe. In Ghana, more religious men experienced negative life-events than non-religious men while in Europe fewer religious men reported negative life-experiences. CVD risk differed significantly between men in Europe as about 66% of non-religious men had elevated risk compared to 54% of religious men. The most frequently reported religious denomination in Ghana was Reformed Christianity (23%) and Pentecostalism in Europe (24%). 

Table 2 illustrates that in Europe more women reported to practise religion (92.2%, *n* = 926) than in Ghana (72.0%, *n* = 598). No women reported smoking in Ghana while about 5% of the non-religious and 1% of the religious women in Europe smoked. None of the women in both Ghana and Europe reported using alcohol. More diabetes cases were detected in religious women in Ghana and Europe. Religious women in Ghana appeared less physically active and experienced more negative life-events than non-religious women. In Ghana, 18% of non-religious women had elevated CVD risk compared to 27% of the religious women. Similarly to men, the majority of women in Ghana were Reformed Christians (26%) and in Europe, Pentecostals (32%). Women tended to visit religious services more often than men (94% women and 92% men in Ghana, and 87% women and 75% men in Europe).

### 3.2. Association between Practising a Religion and Estimated Elevated 10-Year CVD Risk

In men, those practising religion had lower odds of elevated CVD risk than men not practising religion (Table 3). In Europe, men practising religion had 42% lower odds of elevated CVD risk (adjusted odds ratio 0.58; 95% confidence interval 0.39–0.88) compared with men not practising religion when adjusted for education and employment. Additional adjustments for social support, stress, health behaviors and length of stay (0.55; 0.31–0.99) did not meaningfully alter the association. In Ghana, there was tendency for lower odds for elevated CVD risk in those practising religion, but this was not statistically significant. 

In contrast, odds of elevated CVD risk were 53% (1.53; 1.03–2.26) higher for religious women in Ghana compared with non-religious women after adjustment for education and employment. Additional adjustment for social support, stress, and health behaviors did not affect the odds for elevated CVD risk in women (1.53; 1.02–2.30). Among Ghanaian women in Europe, there was no significant difference in the odds of elevated CVD risk for those who did and did not practise a religion (0.90; 0.38–2.13). When the analysis was stratified by the various European sites, practising religion was significantly associated with CVD risk in Berlin only, although the direction of the associations was similar except for London after adjustments for health behaviors.

### 3.3. Association between Religious Denominations and Estimated Elevated 10-Year CVD Risk

Table 4 shows that Ghanaian migrant men in Europe affiliated with Reformed Christianity (0.46; 0.26–0.82), Pentecostalism (0.52; 0.31–0.85), and SDA (0.41; 0.23–0.75) were associated with lower adjusted odds of elevated CVD risk compared with those not practising religion after adjusting for education and employment. Further adjustment for social support, stress, health behaviors and length of stay in Europe resulted in, respectively, 68%, 64%, and 73% lower odds for elevated CVD risk among men affiliated with Roman Catholicism (0.32; 0.14–0.71)**,** SDA (0.36; 0.14–0.88), and other denominations (0.27; 0.08–0.89). Adjusting for health behaviors changed the significance of associations for Roman Catholics and other affiliations reaching statistical significance, which was not the case in earlier models. In Ghana, no association was found between religious denominations and elevated CVD risk for men.

Women in Ghana who practised Reformed Christianity (1.69; 1.03–2.79), SDA (2.15; 1.03–4.52) or other denominations (3.17; 1.42–7.08) had higher odds of elevated estimated CVD risk relative to their non-religious counterparts after controlling for education and employment status. Adjusting for social support, stress and health behaviors did not meaningfully alter associations with an OR of 1.73 (1.03–2.90) for Reformed Christians, 2.15 (1.00–4.62) for SDAs, and 2.81 (1.20–6.54) for other denominations. In Europe, no association was found between religious denominations and elevated CVD risk for women.

### 3.4. Association between Attending Religious Services and Estimated Elevated 10-Year CVD Risk

Table 5 indicates that Ghanaian men residing in Europe who attended religious services had lower odds compared with those not attending religious services. The effect was attenuated after adjustment for social support, stress, health behaviors, and length of stay in Europe for those visiting once a week or more and those visiting once every two weeks or once a month. The odds for elevated CVD risk differed for attending religious services at any frequency compared with not attending religious services among both men and women in Ghana and Europe. 

## 4. Discussion

This study revealed that practising Christian religion is associated with lower odds for elevated CVD risk among Ghanaian men living in Europe but not among men living in Ghana. In contrast, practising Christian religion was associated with higher odds of elevated CVD risk for women living in Ghana but not for women living in Europe. 

The association between religion and lower CVD risk among Ghanaian male migrants, particularly in Berlin, was reported previously in other populations. These studies demonstrated a favourable cardiovascular profile for religiously affiliated persons [12,20,21]. Contrasting to indications from studies in other populations, education, employment, social support, psychosocial stress, and health behaviors did not explain the observed differences among Ghanaians. A study with African Americans indicated that those involved in religious and spiritual activities consumed less alcohol, smoked less, and had a lower energy intake than those less religiously involved [27]. Our results indicate a similar pattern for religious Ghanaian male migrants as they reported to use less alcohol and tobacco compared with those not practising religion. On the contrary, Ghanaian male migrants in Europe who were not practising religion seemed to be more physically active than those who were practising religion. However, adjustment for health-related behaviors including physical activity, alcohol consumption and daily energy intake only altered the observed associations in men from some denominations, such as Roman Catholics. These findings seem to suggest that other (unmeasured) factors may play an important role in the association between religious denominations and CVD risk [17,20].

A striking finding was the higher odds of elevated CVD risk among religious women living in Ghana, particularly in urban centres for which reasons are unclear. The data seem to demonstrate that regulation of health behaviors such as tobacco and alcohol consumption are unlikely to contribute to observed differences in associations between religion and CVD risk as very few Ghanaian women in both Ghana and Europe smoke and consume alcohol. Yet, it is possible that this elevated risk is related to religious fatalism (believing that God, not the individual, has control over health outcomes) [29]. Fatalistic beliefs have shown to be positively associated with self-reported hypertension, diabetes, and hypercholesterolemia [29]. Religious women in Ghana may possibly endorse more fatalistic beliefs and therefore take less responsibility for self-management of CVD risk factors. Fatalistic believes have been shown to be associated with poor medication adherence among diabetes patients [30]. Seeking health care can also be condemned as having too little faith [31]. It is hypothesised that these fatalistic beliefs occur to a lesser extent among Ghanaian women who live in Europe due to exposure to more health education and different health care settings. This may explain the (non-significant) lower odds of elevated CVD risk among women living in Europe compared to Ghana. On the contrary, those experiencing health issues may be more likely to seek for support through religious activities than those without health issues. Our results showed that women practising religion in Ghana reported more experiences of negative life-events than those not practising religion. Further research is needed to test these hypotheses.

### Strengths and Limitations of the Study

The main strengths of the RODAM study are the relatively homogenous study population and the use of well-standardised approaches across sites [9]. The first limitation concerns the cross-sectional design, which does not allow for conclusions that practising religion actually leads to CVD risk. Second, indicators of spirituality were not included meaning that information on one’s self-regulation abilities, and therefore potentially reduction in psychosocial stress, may be missing. Third, socially desirable answers about alcohol and smoking behavior among those religiously involved may possibly have led to an overestimation of the health-behavior related factors, especially among men. Fourth, given its cross-sectional nature, the correlation of estimated ASCVD risk with incident CVD events requires confirmation from prospective studies, as the PCE-risk algorithms used in predicting ASCVD in this study have not yet been validated for sub-Saharan African populations.

## 5. Conclusions

Our findings suggest that Christian religious Ghanaian men living in Europe seem to have lower CVD risk compared with their non-religious counterparts, while Christian religious women in Ghana appear to have increased CVD risk. Further research is needed to gain more insight into the underlying processes and differences between sex and environmental setting in order to guide future CVD prevention programmes for this high-risk population. 

### Key Points

The study found that Christian religious Ghanaian men living in Europe seem to have lower CVD risk compared with their non-religious counterparts, while Christian religious women in Ghana appear to have increased CVD risk.The contributing factors and differences between sex and environmental settings need to be further unravelled.Insight into these underlying processes can guide future CVD prevention programmes for this high-risk population.

## Figures and Tables

**Figure 1 ijerph-18-02451-f001:**
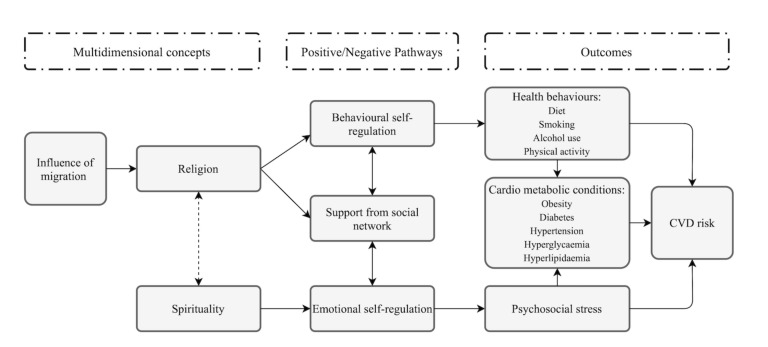
Conceptual framework of potential mechanisms relating religion/spirituality to cardiovascular disease (CVD) risk among African non-migrants and migrants.

**Table 1 ijerph-18-02451-t001:** General characteristics of the male Research on Obesity and Diabetes among African Migrants (RODAM) study population.

Site	Ghana (*n* = 391)		Europe (*n* = 778)	
Men	Not Practising a Religion, (*n* = 118)	Practising a Religion, (*n* = 273)	Not Practising a Religion, (*n* = 129)	Practising a Religion, (*n* = 649)
Age, years ^1^	52.80 (8.11)	53.33 (8.19)	52.42 (6.67)	51.56 (6.97)
Education level, N (%)				
Elementary or less	47 (39.83)	79 (28.94)	16 (12.40)	99 (15.25)
Lower secondary	51 (43.22)	115 (42.12)	56 (43.41)	270 (41.60)
Intermediate	14 (11.86)	52 (19.05)	39 (30.23)	149 (22.96)
Higher schooling/university	6 (5.08)	26 (9.52)	17 (13.18)	124 (19.11)
Employment status, N (%)				
Employed full-time	44 (37.29)	73 (26.74)	80 (62.02)	403 (62.10)
Employed part-time	58 (49.15)	151 (55.31)	14 (10.85)	86 (13.25)
Unemployed/looking for work	3 (2.54)	9 (3.30)	16 (12.40)	58 (8.94)
Other	13 (11.02)	39 (14.29)	18 (13.95)	97 (14.95)
Length of stay in Europe in years ^2^	-	-	22.54 (13.71–27.40)	20.68 (12.66–24.72)
Smoking, N (%) Current	7 (5.93)	8 (2.93)	28 (21.71)	34 (5.24)
Mean systolic blood pressure ^1^	131.19 (18.38)	131.37 (21.87)	141.83 (17.83)	138.91 (17.42)
Antihypertensive medication, N (%)	8 (6.78)	13 (4.76)	29 (22.48)	110 (16.95)
Total cholesterol, mg/dL ^1^	193.17 (40.55)	192.52 (41.24)	203.78 (34.45)	200.42 (37.01)
HDL cholesterol, mg/dL ^1^	48.52 (12.92)	46.91 (11.94)	51.46 (11.61)	51.97 (11.93)
Type-2 diabetes, N (%)	15 (12.71)	27 (9.89)	21 (16.28)	99 (15.26)
Total energy intake (kcal/day) ^2^	2269.59 (1776.29–2787.45)	2380.56 (1940.46–2872.77)	3041.54 (2168.82–3736.59)	2710.12 (1997.10–3608.50)
Alcohol units per day ^2^	0.03 (0–0.29)	0 (0–0.08)	0.18 (0–0.71)	0 (0–0.25)
High-level physical activity, N (%)	77 (65.25)	166 (60.81)	57 (52.78)	260 (47.79)
Psychosocial stress: home and work, N (%)				
Never experienced	34 (28.81)	92 (34.20)	62 (48.06)	333 (51.39)
Permanent stress	0 (0)	6 (2.23)	4 (3.10)	35 (5.40)
Experienced a negative life event in the past 12 months, N (%)	77 (65.25)	189 (69.23)	83 (64.34)	381 (58.71)
SSQT, Social Support Questionnaire for Transactions, sum score 5 items ^2^	15 (14–15)	15 (10–15)	15 (12–17)	15 (13–17)
SSQT, Social Support Questionnaire for Transactions, sum score 5 items ^2^	15 (15–15)	15 (14–15)	15 (14–15)	15 (14–15)
Current religious denomination, N (%)				
Roman Catholic	-	30 (10.99)	-	95 (14.64)
Orthodox Christian	-	35 (12.82)	-	36 (5.55)
Protestant	-	30 (10.99)	-	143 (22.03)
Reformed Christian	-	63 (23.08)	-	92 (14.18)
Pentecostal	-	37 (13.55)	-	155 (23.88)
Seventh Day Adventist	-	25 (9.16)	-	78 (12.02)
Islamic	-	40 (14.65)	-	15 (2.31)
Other	-	13 (4.76)	-	35 (5.39)
Frequency of attending religious services in past 6 months among religious persons, N (%)				
Once a week or more	-	250 (91.58)	-	488 (75.19)
Once every 2 weeks or once a month	-	10 (3.66)	-	95 (14.64)
Less than once a month or never	-	13 (4.76)	-	66 (10.17)
10-year CVD risk, N (%)				
Low risk (<7.5%)	63 (53.39)	156 (57.14)	44 (34.11)	301 (46.38)
Elevated risk (≥7.5%)	55 (46.61)	117 (42.86)	85 (65.89)	348 (53.62)

^1^ Mean + (SD), ^2^ Median + (25th and 75th percentiles).

**Table 2 ijerph-18-02451-t002:** General characteristics of the female RODAM study population.

Site	Ghana (*n* = 831)		Europe (*n* = 1004)	
Women	Not Practising Religion (*n* = 233)	Practising a Religion (*n* = 598)	Not Practising a Religion (*n* = 78)	Practising a Religion (*n* = 926)
Age, years ^1^	51.39 (8.11)	52.47 (8.19)	49.85 (6.29)	50.29 (6.52)
Education level, N (%)				
Elementary or less	148 (63.52)	368 (61.54)	31 (39.74)	274 (29.59)
Lower secondary	72 (30.90)	181 (30.27)	24 (30.77)	329 (35.53)
Intermediate	9 (3.86)	36 (6.02)	18 (23.08)	224 (24.19)
Higher schooling/university	4 (1.72)	13 (2.17)	1 (1.28)	81 (8.75)
Employment status, N (%)				
Employed full-time	71 (30.47)	125 (20.90)	20 (25.64)	340 (36.72)
Employed part-time	143 (61.37)	361 (60.37)	18 (23.08)	256 (27.65)
Unemployed and looking for work	4 (1.72)	24 (4.01)	6 (7.69)	63 (6.80)
Other	15 (6.44)	88 (14.72)	32 (41.03)	247 (26.67)
Length of stay in Europe in years ^2^	-	-	20.21 (14.05–23.71)	20.71 (12.32–26.07)
Smoking, N (%) Current	0 (0)	0 (0)	4 (5.19)	11 (1.19)
Mean systolic Blood pressure ^1^	127.63 (17.83)	128.70 (20.00)	138.21 (16.82)	135.62 (16.61)
Antihypertensive medication, N (%)	10 (4.29)	40 (6.69)	15 (19.23)	155 (16.74)
Total cholesterol, mg/dL ^1^	203.71 (38.0)	207.94 (41.48)	202.93 (39.22)	200.19 (34.12)
HDL cholesterol, mg/dL ^1^	49.44 (11.89)	49.04 (11.64)	58.32 (13.14)	57.68 (12.59)
Type 2 diabetes, N (%)	13 (5.58)	67 (11.21)	7 (8.97)	111 (11.99)
Total energy intake (kcal/day) ^2^	2187.66 (1729.91–2737.55)	2208.81 (1841.14–2733.85)	2544.12 (1664.49–3479.26)	2513.50 (1871.58–3362.51)
Alcohol units per day ^2^	0 (0–0.03)	0 (0–0.02)	0 (0–0.07)	0 (0–0.07)
High-level physical activity, N (%)	137 (58.80)	270 (45.15)	28 (50.0)	346 (45.89)
Psychosocial stress: home&work, N (%)				
Never experienced	69 (30.0)	158 (26.69)	33 (44.59)	447 (49.23)
Permanent stress	1 (0.43)	11 (1.86)	4 (5.41)	36 (3.96)
Experienced a negative life event in the past 12 months, N (%)	119 (51.74)	421 (71.11)	51 (67.11)	546 (59.87)
SSQT sum score 5 items ^2^	15 (15–15)	15 (10–15)	15 (13–16)	15 (14–17)
SSQS, sum score 5 items ^2^	15 (15–15)	15 (14–15)	15 (13–15)	15 (15–15)
Current religious denomination, N (%)				
Roman Catholic	-	62 (10.37)	-	114 (12.31)
Protestant	-	60 (10.03)	-	238 (25.70)
Reformed Christian	-	156 (26.09)	-	112 (12.10)
Pentecostal	-	103 (17.22)	-	293 (31.64)
Seventh Day Adventist	-	45 (7.53)	-	61 (6.59)
Islamic	-	55 (9.20)	-	8 (0.86)
Other	-	32 (5.35)	-	48 (5.18)
Frequency of attending religious services in past 6 months among religious persons, N(%)				
Once a week or more	-	564 (94.31)	-	804 (86.83)
Once every 2 weeks or once a month	-	25 (4.18)	-	81 (8.75)
Less than once a month or never	-	9 (1.51)	-	41 (4.43)
10-year CVD risk, N (%)				
Low risk (<7.5%)	191 (81.97)	436 (72.91)	58 (74.36)	725 (78.29)
Elevated risk (≥7.5%)	42 (18.03)	162 (27.09)	20 (25.64)	201 (21.71)

^1^ Mean + (SD), ^2^ Median + (25th and 75th percentiles).

**Table 3 ijerph-18-02451-t003:** Odds ratios for estimated elevated (≥7.5%) 10-year CVD risk in those practising a religion compared with not practising a religion, stratified by sex and site.

Men: Currently Practising a Religion		OR (95% CI)	
	Crude	Minimally adjusted	Fully adjusted
Ghana	0.86 (0.56–1.33)	0.76 (0.48–1.20)	0.73 (0.44–1.18)
Europe	0.60 (0.40–0.89) *	0.58 (0.39–0.88) *	0.55 (0.31–0.99) *
Women: Currently Practising a Religion		OR (95% CI)	
	Crude	Minimally adjusted	Fully adjusted
Ghana	1.69 (1.16–2.47) **	1.53 (1.03–2.26) *	1.53 (1.02–2.30) *
Europe	0.80 (0.47–1.37)	0.82 (0.47–1.44)	0.90 (0.38–2.13)

Reference category: individuals not practising a religion. Crude model is unadjusted. Minimally adjusted model = adjustment for education and employment. Fully adjusted model = adjusted for education, employment, and SSQT, SSQS (perceived social support), and psychosocial stress, experiences of negative life events (perceived psychosocial stress related factors), and physical activity level, daily total energy intake, daily alcohol consumption (health behaviors), and length of stay in Europe. * *p* < 0.05, ** *p* < 0.01.

**Table 4 ijerph-18-02451-t004:** Odds ratios for estimated elevated (≥7.5%) 10-year CVD risk for religious denominations compared with not practicing a religion, stratified by sex and site.

Men: Religious Denominations		OR (95% CI)	
Ghana	Crude	Minimally adjusted	Fully adjusted
Roman Catholic	1.31 (0.58–2.92)	1.09 (0.46–2.55)	1.07 (0.44–2.60)
Orthodox Christian	0.86 (0.40–1.84)	0.71 (0.31–1.62)	0.62 (0.26–1.49)
Protestant	0.76 (0.34–1.73)	0.64 (0.27–1.53)	0.65 (0.27–1.59)
Reformed Christian	0.98 (0.53–1.80)	0.86 (0.45–1.64)	0.77 (0.39–1.52)
Pentecostal	0.70 (0.33–1.49)	0.66 (0.29–1.46)	0.61 (0.27–1.38)
Seventh Day Adventist	0.76 (0.32–1.84)	0.65 (0.26–1.66)	0.65 (0.24–1.80)
Islamic	0.69 (0.33–1.43)	0.67 (0.31–1.47)	0.75 (0.33–1.73)
Other	0.98 (0.31–3.10)	0.85 (0.25–2.82)	0.76 (0.22–2.65)
Europe	Crude	Minimally adjusted	Fully adjusted
Roman Catholic	0.68 (0.40–1.18)	0.65 (0.37–1.13)	0.32 (0.14–0.71) **
Orthodox Christian	0.46 (0.22–0.98) *	0.49 (0.23–1.06)	0.71 (0.26–1.92)
Protestant	1.02 (0.62–1.69)	1.02 (0.61–1.71)	0.97 (0.45–2.05)
Reformed Christian	0.52 (0.30–0.89) *	0.46 (0.26–0.82) *	0.55 (0.24–1.23)
Pentecostal	0.51 (0.32–0.83) **	0.52 (0.31–0.85) *	0.73 (0.35–1.53)
Seventh Day Adventist	0.42 (0.24–0.75) **	0.41 (0.23–0.75) **	0.36 (0.14–0.88) *
Islamic	0.35 (0.12–1.03)	0.33 (0.11–1.01)	0.35 (0.08–1.61)
Other	0.55 (0.26–1.17)	0.52 (0.24–1.12)	0.27 (0.08–0.89) *
**Women: Religious Denominations**		**OR (95% CI)**	
Ghana	Crude	Minimally adjusted	Fully adjusted
Roman Catholic	1.09 (0.53–2.23)	0.94 (0.45–1.96)	1.06 (0.49–2.28)
Orthodox Christian	1.79 (1.00–3.19) *	1.56 (0.85–2.85)	1.49 (0.79–2.82)
Protestant	1.95 (1.02–3.72) *	1.68 (0.85–3.29)	1.55 (0.77–3.14)
Reformed Christian	1.73 (1.07–2.81) *	1.69 (1.03–2.79) *	1.73 (1.03–2.90) *
Pentecostal	1.70 (0.98–2.94)	1.59 (0.90–2.80)	1.60 (0.89–2.86)
Seventh Day Adventist	2.05 (1.01–4.19) *	2.15 (1.03–4.52) *	2.15 (1.00–4.62)
Islamic	0.89 (0.40–1.96)	0.57 (0.25–1.31)	0.54 (0.22–1.32)
Other	3.54 (1.63–7.67) *	3.17 (1.42–7.08) **	2.81 (1.20–6.54) *
Europe	Crude	Minimally adjusted	Fully adjusted
Roman Catholic	0.86 (0.44–1.68)	0.99 (0.44–1.78)	0.72 (0.24–2.14)
Orthodox Christian	0.78 (0.34–1.80)	0.77 (0.31–1.87)	0.87 (0.24–3.12)
Protestant	0.71 (0.39–1.30)	0.71 (0.38–1.33)	0.83 (0.33–2.13)
Reformed Christian	1.16 (0.60–2.23)	1.07 (0.54–2.12)	1.61 (0.58–4.50)
Pentecostal	0.76 (0.43–1.36)	0.82 (0.44–1.52)	1.01 (0.39–2.62)
Seventh Day Adventist	0.64 (0.28–1.46)	0.69 (0.29–1.67)	0.52 (0.13–2.16)
Islamic	0.41 (0.05–3.58)	0.47 (0.05–4.32)	-
Other	0.97 (4.2–2.21)	0.87 (0.36–2.11)	0.55 (0.13–2.28)

Reference category: individuals not practising a religion. Crude model is unadjusted. Minimally adjusted model: adjustment for education and employment. Fully adjusted model: adjusted for education, employment, and SSQT, SSQS (perceived social support), and psychosocial stress, experiences of negative life events (perceived psychosocial stress related factors), and physical activity level, daily total energy intake, daily alcohol consumption (health behaviors), and length of stay in Europe. * *p* < 0.05, ** *p* < 0.01.

**Table 5 ijerph-18-02451-t005:** Odds ratios for estimated elevated (≥7.5%) 10-year CVD risk for frequency of attending religious services compared with not attending religious services, stratified for sex and site.

Men: Attending Religious Services		OR (95% CI)	
Ghana	Crude	Minimally adjusted	Fully adjusted
Once a week or more	1.74 (0.52–5.80)	1.88 (0.49–7.27)	1.69 (0.42–6.71)
Once every two weeks or once a month	1.50 (0.27–8.45)	1.74 (0.26–11.56)	1.27 (0.18–8.79)
Europe	Crude	Minimally adjusted	Fully adjusted
Once a week or more	0.47 (0.27–0.81) *	0.44 (0.25–0.78) **	0.75 (0.35–1.62)
Once every two weeks or once a month	0.46 (0.24–0.90) *	0.45 (0.23–0.90) *	0.58 (0.23–1.44)
**Women: Attending Religious Services**		**OR (95% CI)**	
Ghana	Crude	Minimally adjusted	Fully adjusted
Once a week or more	1.34 (0.28–6.51)	1.49 (0.29–7.51)	1.74 (0.31–9.95)
Once every two weeks or once a month	0.67 (0.01–4.46)	0.73 (0.10–5.09)	1.01 (0.13–7.93)
Europe	Crude	Minimally adjusted	Fully adjusted
Once a week or more	1.37 (0.60–3.15)	2.11 (0.84–5.26)	3.37 (0.69–16.60)
Once every two weeks or once a month	1.29 (0.49–3.42)	2.45 (0.85–7.09)	2.28 (0.38–13.74)

Reference category: individuals attending religious services less than once a month or never. Crude model is unadjusted. Minimally adjusted model: adjustment for education and employment. Fully adjusted model: adjusted for education, employment, and SSQT, SSQS (perceived social support), and psychosocial stress, experiences of negative life events (perceived psychosocial stress related factors), and physical activity level, daily total energy intake, daily alcohol consumption (health behaviors), and length of stay in Europe. * *p* < 0.05, ** *p* < 0.01.

## Data Availability

The datasets generated and/or analyzed during the current study are not publicly available due to information that could compromise research participant privacy but are available from the corresponding author on reasonable request.

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
