# Peer review of "Association between Practising Religion and Cardiovascular Disease Risk among Ghanaian Non-Migrants and Migrants in Europe: The RODAM Study"

_ijerph, 2021, doi:10.3390/ijerph18052451_

Round 1
Reviewer 1 Report
Overall this is a well-written paper with an interesting and novel result.
INTRODUCTION
The introduction provides enough background information for readers to understand the problem. The introduction provides a good perspective on the main topic.
Motivations for this study are more than clear. Please include a concise study objective and hypothesis.
METHODS
Well described
RESULTS
Results paragraph should include more relevant and extended data.
All of the tables include specific and well-developed statistic.
The author presented a good introduction, methods and results.
DISCUSSION
the role of physical activity or medication consumption must be discussed
Include the Limitations of the study.
The main aim to improve the paper is to focus on the discussion. There are some factors like physical activity and the consumption of medication that must be discussed to improve the quality of the paper.
LITERATURE CITED
The literature cited is relevant to the study
SIGNIFICANCE AND NOVELTY
As it stands, the results are novel and important enough for this journal.
In addition, a concise objective is needed as well as a conclusion that response to this objective.
The author presented a good introduction, methods and results. The main aim to improve the paper is to focus on the discussion. There are some factors like physical activity and the consumption of medication that must be discussed to improve the quality of the paper. In addition, a concise objective is needed as well as a conclusion that response to this objective.
Reviewer 2 Report
The article is interesting and valuable. I believe that the study carried out is purposeful. The introduction state the purpose of the paper. The number of positions in the references section is proper and references are modern.The tables are high quality. The article is clearly written. The Ghanaian research group is impressive. A large-scale experiment was performed. The conducted statistical research is elementary. STATA version 14.2 was used to analyze the statistical data. The assumptions of the article have been fulfilled. The discussion is critical and polemical to the research carried out. An interesting part of the article is conducting research on immigrants in European cities. Relevant observations were presented.The research was conducted in accordance with the ethical assumptions.
Mandatory change:
1. The authors should add Conclusions section to the article. This section should contains important summary of the article and for example direction of next investigations.
2. You should increase the resolution of figure 1
Reviewer 3 Report
The paper describes the investigation of the possible association between religious beliefs and cardiovascular disease risk among Ghanaian non-migrants and migrants in Europe using retrospective data from a cross-sectional study.
The paper would be indeed timely and of interest for the journal readers, however I believe the analyses do not address the problem adequately. In particular, the method used to investigate the proposed model is not appropriate for the theoretical assumptions stated.
1) Reporting the data: In the statistical methods section, the authors state that they used logistic regression, but in the tables, only the adjusted ORs are reported. My understanding given the notation used is that they used a multivariate logistic regression and reported only the adjusted results. However, in order to correctly report the results of a multivariate regression, the univariate estimated ORs and the p-values should be reported first. These results help readers understand the need of a multivariate analysis and, on the basis of these results, it is possible to decide which variables to include, then, in a multivariate model. Additionally, it is usual to include in a multivariate analysis only variables that are significantly associated in the univariate analysis.
2) Statistical analysis choice: Regardless of how data are reported, a multivariate logistic regression does not address the study problem satisfyingly.
I do understand that the aim of the authors was to test for confounding variables (and thus a multivariate logistic regression would have been a good fit), however, in the image of the theoretical causal inference model proposed, the effect between religious beliefs and CVD-risk is not modified by confounding variables but by a chain of mediator variables (and moderator variables), therefore the adjusted odds ratios of a multivariate regression will underestimate the effect of the predictors and thus cannot be just interpreted as they are. A mediator analysis should be carried out to investigate the true effect of religious belief alone and the impact of all the mediators and moderators involved investigating, for example, the residuals. Another approach that would be even more appropriate in understanding the complexity of these relations could be a path model approach.
Since the statistical analysis is not appropriate, all the results drawn from the study cannot be verified.
Round 2
Reviewer 3 Report
The model remains inadequate as well as the statistical analyses (now the p-values are missing). I believe the paper needs to be completely rewritten in light of appropriate statistical analyses. I recommend to submit your paper to a statistician.
Author Response
We have provided effect estimates with their corresponding 95% CIs, which are more robust and an average scientist prefers this and is able to interpret 95% CIs whether it is significant or not. In fact in most journals, it now a requirement to report effect estimates with corresponding 95% CIs rather than p-vales. Nevertheless, we have also indicated where the effect estimates and their corresponding 95% CIs are significant with asterisk for those who are unable to interpret 95% CIs and the significant levels are indicated in footnotes.